A morpho-histological analysis of the exoskeleton of Clathrozoella medeae (Cnidaria, Hydrozoa) reveals insights into the taxonomy of Clathrozoellidae and Hydroidolina

Mendoza-Becerril María A. m_angelesmb@hotmail.com 1
Marques Antonio C. 2 3
1 Department of Aquatic Systematics and Ecology, El Colegio de la Frontera Sur-Chetumal (ECOSUR) , Chetumal , Quintana Roo , Mexico
2 Department of Zoology, Institute of Biosciences, University of São Paulo , São Paulo , Brazil
3 NOAA Fellow in Marine Science, National Museum of Natural History (NMNH) , Washington D.C. , United States of America
Idris Izwandy
Electronic publication date: 2025 Jan 9
Publication date: 2025
Volume: 13
Electronic Location ID: e18407
Received 2024 Aug 30; Accepted 2024 Oct 5
Copyright: ©2025 Mendoza-Becerril and Marques
Copyright year: 2025
Copyright holder: Mendoza-Becerril and Marques
License: This is an open access article distributed under the terms of the Creative Commons Attribution License, which permits unrestricted use, distribution, reproduction and adaptation in any medium and for any purpose provided that it is properly attributed. For attribution, the original author(s), title, publication source (PeerJ) and either DOI or URL of the article must be cited.
License URL: https://creativecommons.org/licenses/by/4.0/

Keywords: Antarctica, Anthoathecate, Histochemistry, Polyp, Pseudohydrotheca

Funding: CAPES/CNPq –IEL National –Brazil Proc. 6101100-2011 São Paulo Research Foundation (FAPESP) CNPq 316095/2021-4 Proc. 2010/52324-6 2010/06927-0 2011/50242-5 2023/17191-5 This study was supported by a fellowship from CAPES/CNPq –IEL National –Brazil (Proc. 6101100-2011) for María A. Mendoza-Becerril. Additional funding was provided by CNPq 316095/2021-4 and São Paulo Research Foundation (FAPESP) Proc. 2010/52324-6, 2010/06927-0, 2011/50242-5, 2023/17191-5 for Antonio C. Marques. The funders had no role in study design, data collection and analysis, decision to publish, or preparation of the manuscript.

==============================
The taxonomic complexity of the families Clathrozoidae and Clathrozoellidae, rooted in early 20th-century hydroid descriptions, highlights the need for comprehensive and detailed morphological analyses. This study aimed to elucidate the histology of the polypoid stage of Clathrozoella medeae Peña Cantero, Vervoort & Watson, 2003, with a particular emphasis on its exoskeletal structure. Specimens from the National Museum of Natural History were examined histologically using different staining techniques. The results revealed a three-layered mesoglea, diverse gland cells, and an exoskeleton comprising chitin and structural proteins, with notable differences from other anthoathecate hydroids. These results have significant implications for the taxonomy and evolutionary relationships of Clathrozoellidae and Hydroidolina, as they highlight the importance of detailed histological data in understanding the unique exoskeletal architecture of C. medeae, termed “exoskeleton tube”, which distinguishes it from other hydroids, and provide critical insights into the homology and phylogenetic position of Clathrozoellidae.

Introduction

The taxonomic history of the families Clathrozoidae (Stechow, 1921) and Clathrozoellidae Peña Cantero, Vervoort & Watson, 2003 is complex. It began with the description of the new genus and species Clathrozoon wilsoni Spencer, 1891, based on material from near Port Phillip Heads, Victoria (Australia). Spencer (1891) highlighted the unique morphology of the new taxon, comparing it to “Anthoathecata” (viz., Hydractiniidae and Solanderiidae, as “Ceratelladae” and “Hydrocorallinae”) because of the similarity of the exoskeleton, and to Leptothecata (viz., Plumulariidae) because of the presence of nematothecae. He placed his new species and new genus in the new family Hydroceratinidae Spencer, 1891, on the basis of a “combination of characters, [that] together with the nature of the skeleton, serves to render the Hydroceratinidae distinct from any family of Hydroidea yet known” (Spencer, 1891: 129). Although Spencer (1891) used the term “hydrothecae” to describe the morphology of his new species, he did not establish anthoathecate or leptothecate assignments for Hydroceratinidae, contrary to the observation by Vervoort & Watson (1996: 119).

Clathrozoon was not recorded again until Vanhöffen (1910) described a second species for the genus, on the basis of material from the Davis Sea (Antarctica, 385 m deep), and named it “Clathrozoon Drygalskii” Vanhöffen, 1910. Vanhöffen clearly placed his new species among the anthoathecates, still within the family Hydroceratinidae (Vanhöffen, 1910: 291), despite the generic use of polyp tubes (as “Polypenröhren”, in German, Vanhöffen, 1910: 294). However, the family name Hydroceratinidae is known to be incorrect because it is not based on an existing genus (WoRMS, 2024).

Subsequently, Stechow (1921) observed differences between the exoskeletons of C. wilsoni and C. drygalskii and assigned Vanhöffen’s species as the type species of his new genus Clathrozoella Stechow, 1921. He also noted similarities between both species with the anthoathecate genera Nuttingia and Hydrodendrium (currently Hydractinia) and the leptothecate genus Keratosum (presently Lafoeina), suggesting that they form the subfamily Clathrozoinae (Stechow, 1921) or the family Clathrozoidae (Stechow, 1921) (cf. Stechow, 1921; see also Crowell, 1982 on Stechow’s comments of regarding the uncertain position of Keratosum). Stechow used the term pseudotheca (in German, “pseudotheken”) to describe the morphology of the species and retained the new family among the anthoathecates (Stechow, 1921).

Hirohito (1967) proposed the new genus Pseudoclathrozoon for a species related to Clathrozoon, considering both as leptothecate hydroids. In the same study, Hirohito explicitly excluded Clathrozoella from Clathrozoidae and considered its affinity with Leptothecata uncertain. This position was later confirmed by Vervoort & Watson (1996), who stated Clathrozoella had an uncertain affinity with “Anthoathecata”, either with “Filifera” Hydractiniidae or Capitata Solanderiidae. Indeed, Vervoort (2000: 239) described the presence of unprotected developing female gonophores next to the hydranth body, communicating with the coenosarc of the tubules, as well as “desmones” (referring to the desmonemes), both characters expected in an anthoathecate representative (see also Peña Cantero, Vervoort & Watson, 2003; Calder, Choong & McDaniel, 2015). However, he considered Clathrozoidae, including C. wilsoni, to have a “false hydrotheca” (Vervoort, 2000: 237).

Clathrozoella remained monospecific until Peña Cantero, Vervoort & Watson (2003) described three new species, namely Clathrozoella abyssalis, Clathrozoella bathyalis, and Clathrozoella medeae. These authors proposed the new family Clathrozoellidae, agreeing with its anthoathecate affinity and following Stechow (1921) in the use of the term “pseudohydrothecae”, confirming it as a structure distinct from Leptothecatae hydrotheca (Peña Cantero, Vervoort & Watson, 2003: 282). Meanwhile, Clathrozoidae was maintained as a separate and valid family, still assigned to Leptothecata, including Clathrozoon wilsoni Spencer, 1891 and Pseudoclathrozoon cryptolarioides Hirohito, 1967.

The taxonomic history of Clathrozoellidae highlights the benefits of broader and more integrated data in systematics, including gonophore and exoskeletal morphology, cnidome, and sequence data (Calder, Choong & McDaniel, 2015). The exoskeleton is a key morphological character for Clathrozoellidae and Clathrozoidae. However, the variety of terms used in the studies, such as “hydrotheca” (e.g., Spencer, 1891), “polyp tubes” (as “Polypenrohren”, Vanhöffen, 1910), and “Pseudotheken” or “false hydrotheca” or “pseudohydrothecae” (e.g., Stechow, 1921; Vervoort, 2000; Peña Cantero, Vervoort & Watson, 2003; respectively), indicate uncertainties in defining homologies and understanding the evolutionary relationships of these groups.

Interestingly, the “pseudohydrothecae” structure mentioned above differs from the homonym described in other anthoathecate hydroids (cf. Mendoza-Becerril et al., 2017). Although the literature contains preliminary histological data on C. drygalskii (Vervoort, 2000), further histological studies on the tissue organization and chemical affinities of Clathrozoella, including the nature of the exoskeletons, would improve be our understanding of the group’s affinities and exoskeletal architecture among hydroids (i.e., Leptothecata and the non-monophyletic “Anthoathecata”; see Cartwright et al., 2008; Maronna et al., 2016; Mendoza-Becerril et al., 2018). Unfortunately, this lack of knowledge is not limited to these taxa—histological studies on hydroids are rare and generally focused on freshwater Hydra (e.g., Parker, 1879; Siebert, Anton-Erxleben & Bosch, 2008), with few studies on Leptothecata (e.g., Shimabukuro & Marques, 2006; Pyataeva & Kosevich, 2008) and “anthoathecate” non-calcareous polyps (e.g., Warren, 1907; Wineera, 1968; Wineera, 1972; Mendoza-Becerril et al., 2016; Mendoza-Becerril et al., 2017).

The aim of this study is to describe the histology of the polypoid stage, including the exoskeleton, of Clathrozoella medeae Peña Cantero, Vervoort, & Watson, 2003 a species endemic to Antarctica (Peña Cantero, Vervoort & Watson, 2003; Miranda et al., 2021). These data will be used to understand the taxonomic implications of the nature and organization of the exoskeleton for the group and hydroids in general.

Material & Methods

The material studied is part of the collection of the National Museum of Natural History, Smithsonian Institution, catalogue number USNM1003100. Collected in the Antarctic, South Shetland Islands (61°24.9′S, 56°30.1′W), at a depth of 300 m on March 13, 1964, by the Department of Zoology from the University of Southern California, the material consists of a colony of polyps with gonophores attached to a rock, preserved in ethanol.

Fragments of the colony, including polyps and parts of the exoskeleton, were dehydrated and embedded in glycol methacrylate (GMA) resin (Leica Historesin Embedding Kit, Leica Microsystems Nussloch GmbH, Germany). Serial longitudinal and transverse sections (3 µm and 7 µm, respectively) of the exoskeleton were stained with different methods: toluidine blue (TB), hematoxylin and eosin (HE), periodic acid–Schiff (PAS, for identification of polysaccharides—P), alcian blue at pH 2.5 (AB, for identification of glycosaminoglycans—GAGs), mercuric bromophenol blue, and naphthol yellow S (HgBPB and NYS, respectively, for identification of proteins) (McManus, 1946; Deitch, 1955; Mowry, 1963; Pearse, 1985). Staining methods (AB + PAS + H) and general staining procedures and times in GMA resin were combined as follows for histological analyses as described by Mendoza-Becerril et al. (2017).

The cnidome was also examined using the nematocyst terminology of Mariscal (1974). The histological slides produced were analyzed using a compound microscope Zeiss Axio Imager M2 and deposited in the collection of the National Museum of Natural History, Smithsonian Institution, under the same catalogue number as the material studied.

Results

The analysis of the longitudinal sections of C. medeae polyps revealed three morphologically distinct regions (Fig. 1A), viz., (a) the hypostome, characterized by well-developed gland cells in the gastrodermal layer; (b) the median body region, containing large vacuolated endodermal cells; and (c) the base of the polyp with gland cells in the epidermal layer.

Figure 1 Morphology and histology of the polyp of Clathrozoella medeae.

(A) Schematic representation of the polyp with three regions: (a) hypostome, (b) middle region, (c) base. (B) Epitheliomuscular cells of the epidermis. (C) Epidermis of the polyp base. (D) Detail of the hypostomal epidermis. (E) Epidermis of a tentacle. (F) Detail of a tentacular epidermis. (G) Detail of the three layers of mesoglea. (H) Mesoglea exhibiting PAS-positivity. (I) Mesoglea in the tentacles. (J) Detail of the gastrodermis featuring gland cell type gcIII. (K) Gastrodermis of the hypostome showing gland cell types gcI, gcII, and gcIII. (L) Detail of a tentacular gastrodermis. Black arrowhead, gcI; blue arrowhead, gcII; orange arrowhead, gcIII; white arrowhead, nematocysts. Abbreviations: ep, epidermis; gc, gland cell; gt, gastrodermis; ms, mesoglea; msI, msII, msIII, three layers of mesoglea; t, tentacle. Scale bars: B, D, E, G, I—10 µm; C—50 µm; F, K—200 µm; H—1.0 mm; J, L—500 µm. Stain: B, D–G, I—HE; C, H, L—PAS; J—AB; K—AB + PAS + H.

The epidermal layer consists of muscular epithelial cells with sinuous surfaces and heterogeneous sizes (Fig. 1B). These cells are vacuolated, with some granulated glandular cells (PAS-positive) showing a higher affinity for HgBPB and NYS (Table 1). Gland cells are more prominent in the basal part of the polyp, generally in the epidermis of the median and basal regions (Figs. 1B and 1C). Nematocysts are scarce in the median and basal regions but abundant in the hypostome (Fig. 1D) and tentacles (Fig. 1E). The most common cell type is the epitheliomuscular cell, which is thinner and presents a linear surface in the hypostome region (Fig. 1D). The tentacle epidermis is more uniform, with cubic cells containing a basal nucleus (Fig. 1F).

The mesoglea is acellular, prominent, and located immediately beneath the epidermis, projecting into folds at the base of the gastrodermis (Fig. 1G). The mesoglea was stained with HE, TB, and PAS (Table 1), and it is divided it into three main parallel layers, defined as msI, msII, and msIII. The msI layer is densely stained with HE and TB and shows a strong PAS-positive reaction. This layer is located inside in the fibers (Fig. 1G). The other two mesogleal layers, msII and msIII, are more external and less densely stained (Fig. 1G). In the tentacles and hypostome, only the msII layer is observed (Figs. 1H and 1I), corresponding to a thin median layer.

The gastrodermis consists mainly of epitheliomuscular and gland cells (Fig. 1F). The hypostome contains two types of gland cells, both stained with the PAS procedure (Fig. 1H). One type, located distally in the oral region, stains intensely with AB (Figs. 1J and 1K), suggesting the presence of acidic GAGs. In addition, the solid tentacles have a core of vacuolated gastrodermal cells and some granular cells (Figs. 1H and 1L).

The stem consists of coenosarcal tubes surrounded by an exoskeleton, with long and curved exoskeletal elements, named “exoskeleton tubes”, arising irregularly around the stem (Figs. 2A–2C). The coenosarcal epidermis contains vacuolated cells, separated from the gastrodermis by a thin, unstructured layer of mesoglea (Figs. 2D–2F), which stained pink with HE (Fig. 2G). Large granulated gland cells, PAS-positive stained with HgBPB and NYS, indicate the presence of proteins related to exoskeletal secretion by epidermal gland cells (Fig. 2H). In the gastrodermis, these cells have smaller globules (Fig. 2G).

Table 1 Reactions of the polyp and exoskeletal layers of to specific staining Clathrozoella medeae Peña Cantero, Vervoort & Watson, 2003.

Structure	TB	HE	Schiff	PAS	AB	HgBPB	NYS	
Polyp								
Epidermis	++ blue	+++ pink	<+ magenta	++ magenta	–	+ blue	++ yellow	
Mesoglea	++ purple	+ purple	<+ magenta	+++ magenta	<+ alcian blue	+ blue	+ yellow	
Gastrodermis	++ blue	++ purple	–	+++ magenta	+ alcian blue	++ blue	+++ yellow	
Cnidome	+++ purple	+++ purple	<+ magenta	+++ purple	+++ alcian blue	+++ blue	++ yellow	
Exoskeletal tube								
Epidermis	++ blue	+++ pink	–	+ magenta	–	+ blue	+ yellow	
Mesoglea	+++ blue	+ pink	–	++ magenta	–	–	–	
Gastrodermis	+++ purple	++ pink	–	+++ magenta	–	++ blue	++ yellow	
Cnidome	+++ purple	++ purple	<+ magenta	++ magenta	+++ alcian blue	+++ blue	++ yellow	
Exoskeleton								
Inner layer (=perisarc)	+++ blue	+++ pink	<+ magenta	+++ magenta	<+ alcian blue	+++ blue	+ yellow	
Outer layer (=exosarc)	+++ purple	+ pink	–	++ magenta	+++ alcian blue	–	<+ yellow	
Notes.

- not stained

<+ nearly unstained

+ weakly stained

++ moderately stained

+++ intensely stained

Stain: TB, Toluidine blue; HE, hematoxylin and eosin; Schiff, Schiff reagent applied without any pretreatment; PAS, Periodic Acid-Schiff; AB, Alcian blue pH 2.5; HgBPB, mercury-bromophenol blue; NYS, Naphtol yellow S.

Figure 2 Perisarc and coenosarc of the tubes of the stem and nematocysts.

(A–C) General schematic representation with details of the stem. (A) General overview of the stem. (B) Transversal section. (C) Longitudinal section. (D) Transversal section of skeletal tubes. (E–F) Detailed views of panel D. (G) Layers of coenosarc. (H) Detail of the epidermis featuring type gcI gland cells. (I) Coenosarc of the nematophore. (J) Heterotrichous microbasic eurytele and microbasic mastigophores nematocysts in the nematophore. (K) Desmoneme and heterotrichous microbasic eurytele nematocysts in the tentacle. (L) Hypostome with heterotrichous microbasic eurytele nematocysts. (M–N) Coenosarc of the exoskeleton tubes with heterotrichous microbasic eurytele (only in M) and microbasic mastigophore nematocysts. Black arrowhead, gcI; white arrowhead, nematocysts. Abbreviations: cn, coenosarc; d, desmoneme nematocyst; ep, epidermis; et, exoskeleton tubes; gc, gland cell; gt, gastrodermis; hme, heterotrichous microbasic eurytele nematocyst; mm, microbasic mastigophore nematocyst; ms, mesoglea; n, nematotheca; p, polyp. Scale bars: D, H—500 µm; E–G—200 µm; I—25 µm; J—25 µm; K–N—10 µm. Stain: D–F, M—TB; G–I—AB + PAS + H; J—PAS; K–L, N—HE.

The nematophore is enclosed in a perisarcal tube, formed by a pedicelar structure of by epidermis and gastrodermis, ending in a bundle of heterotrichous microbasic eurytele and microbasic mastigophore nematocysts (Figs. 2I and 2J). Nematocysts are PAS-positive and show a strong affinity for AB dye and a moderate affinity for HgBPB and NYS, especially those of the exoskeletal tube (Table 1). Three types of nematocysts were identified, viz., heterotrichous microbasic eurytele (hme), desmoneme (d), and microbasic mastigophore (mm).

Undischarged heterotrichous microbasic euryteles stained with AB + PAS + H have magenta capsules with light purple spines and blue tubules (Figs. 2I and 2J); discharged nematocysts are generally light purple. Heterotrichous microbasic euryteles are present in the epidermis of the nematophore (25.4 × 9.2 µm) (Fig. 2J), tentacles (8.0 × 6.0 µm) (Fig. 2K), and hypostome (8.0 × 4.0 µm) (Fig. 2L), as well as isolated in the coenosarcal epidermis of the exoskeleton tubes (24.0 × 10.0 µm) (Fig. 2M). Desmonemes undischarged capsules, abundant in the apical part of the tentacles, measure 4.0 × 2.0 µm, are PAS-positive compared to the Schiff-control, and stain purple with AB + PAS + H (Fig. 2K). Microbasic mastigophores are present in the coenosarc of the exoskeleton tubes (53.6 × 14.6 µm) (Figs. 2M and 2N).

The exoskeleton consists of a main stem, from which surface tubes arise around the polyps, as well as nematophores surrounded by small tubes. The exoskeleton is two-layers, with an outer layer (exosarc) and an inner layer (perisarc). The exosarc is thin and irregularly shaped (Figs. 3A–3H), extending from the base (18.1 µm thick) of the stem to the nematophore (nematotheca) (0.26 µm thick), composed of GAGs (Table 1) with abundant inorganic and few organic materials (Figs. 3B–3D). Detection of the exosarc in some stem regions, such as in the inner wall of the exoskeleton tube (Fig. 3B), is challenging. The perisarc (inner layer), which has a strong affinity for PAS (Figs. 3D and 3E), is in direct contact with the coenosarcal epidermis. The perisarc is homogeneous and extends from the base of the stem (31.81 µm thick) to the nematotheca (22.72 µm thick) (Figs. 3A and 3F). The stem consists of dense tubes of chitin and structural proteins (Table 1) (Figs. 3G and 3H), with a core of coenosarc with gland cells with an affinity for HgBPB and NYS.

Figure 3 Exoskeleton.

(A) Schematic representation of the exoskeleton: (a) transversal view section, (b) longitudinal view section, (c) nematotheca. (B) Layers of the exoskeleton of the stem. (C) Detail of the exosarc. (D) Detail of the exosarcal and perisarcal layers. (E) Perisarc and type gcI gland cells. (F) Detail of the nematothecal perisarc and exoskeleton tube. (G–H) Perisarc and type gcI gland cells. Black arrowhead, gcI; white arrowhead, nematocysts. Abbreviations: cn, coenosarc; et, exoskeleton tubes; ex, exosarc; gc, gland cell; n, nematotheca; pe, perisarc; st, stem. Scale equals: B, C, F—100 µm; D, E, G, H—50 µm. Stain: B, F—TB; C—AB; D—AB + PAS + H; E—PAS; G—HgBPB; H—NYS.

Discussion

The data highlights the importance of detailed morpho-histological and histochemical information on the exoskeleton for hypothesizing homology within Hydroidolina, particularly between Clathrozoellidae and other families. These characters are essential for inferring phylogenetic relationships or independently testing higher-level taxonomic proposals derived from molecular sequences. Observations on detailed histological analysis have demonstrated that it has phylogenetic implications in medusozoans (e.g., Siebert et al., 2009; García-Rodríguez et al., 2023), as well as other taxonomic data, such as, morpho-molecular and fluorescence patterns (Maggioni et al., 2020; Beckmann et al., 2024).

The general tissue and cellular organization of C. medeae is similar to that described for other anthoathecates, such as Parawrightia robusta (Warren, 1907), Solanderia spp. (Wineera, 1968), Coryne eximia (Wineera, 1972), and bougainvilliids (Mendoza-Becerril et al., 2017). However, notable features include muscular epithelial cells with sinuous surfaces, the three-layered mesoglea, and exoskeletal structure.

The arrangement of the anastomosed coenosarcal tubes is similar to that of Solanderia misakinensis (Wineera, 1968). Historically, affinities between Clathrozoidae or Clathrozoellidae and Solanderiidae have been proposed since the original descriptions by Spencer (1891), Vanhöffen (1910), and more recently by Vervoort & Watson (1996). However, the arrangement of exoskeletal elements in C. medeae differs from all other Hydroidolina, including Solanderiidae, in that they provide support and protection to the entire hydranth. Although Solanderia spp. also have a rigid chitinous skeleton, it is arranged as an internal network of longitudinal and transverse connecting fibers (Wineera, 1968), unlike the external tubes in C. medeae (see Peña Cantero, Vervoort & Watson, 2003: Figs. 4, 5D and 6D). In addition, the exosarc of Clathrozoellidae, recognized since early descriptions as a thick layer of foreign bodies of tiny algae and diatoms (Vervoort & Watson, 1996), contrasts with the external soft layer of S. misakinensis (Wineera, 1968), corresponding to the ectoderm. This suggests that exosarc origin may differ in different anthoathecate clades. Furthermore, molecular analysis using the mitochondrial 16S marker confirmed the affinities of C. drygalskii between “Anthoathecata” and “Filifera”, but not Capitata, and identified it as sister group of Similiclavidae (Calder, Choong & McDaniel, 2015), within a more inclusive clade also including ten other species of Eudendriidae (Calder, Choong & McDaniel, 2015).

A brief terminological discussion is necessary to avoid confusion regarding the nature of exoskeletons. Few polypoid stages of Hydroidolina are completely naked; most possess an exoskeleton. In the non-monophyletic “Anthoathecata”, the exoskeleton is present in the hydrorhiza or in both the hydrorhiza and hydrocaulus, enclosing stolons and coenosarc, but the hydranth is usually naked (cf., Millard, 1975). However, some anthoathecate taxa have a chitinous perisarc and exosarc composed of acid GAGs covering the colony, sometimes forming a pseudohydrotheca when both layers cover the base of the hydranth (Mendoza-Becerril et al., 2016; Mendoza-Becerril et al., 2017).

The exoskeleton of Leptothecata is formed by a continuous layer of chitin and structural proteins, while some anthoathecates may have an exosarc as an additional layer (Mendoza-Becerril et al., 2017). Conversely, the exoskeleton of C. medeae consists of a network of chitin and structural proteins complemented by a thin exosarc layer, both secreted by the ectoderm. The morphology and histology of this exoskeletal structure does not correspond to the pseudohydrotheca found in other hydroids, such as bougainvilliids, which are formed by a corneus chitin-protein reinforced by a covering exosarc formed of GAGs (Mendoza-Becerril et al., 2016). Therefore, the term “exoskeleton tube” is more appropriate for the exoskeleton of C. medeae.

Conclusions

We demonstrated that detailed morpho-histological analysis of the exoskeleton is a useful tool for hypothesizing homology within Hydroidolina, particularly between Clathrozoellidae and other families. The tissue and cellular organization of C. medeae shares similarities with other anthoathecates but has unique elements such as muscular epithelial cells with sinuous surfaces, a three-layer mesoglea, and a distinctive exoskeletal structure. The exoskeleton of C. medeae, which provides support and protection to the entire hydranth, differs from the internal fiber network of Solanderia spp. This suggests that the exosarc may have originated in different anthoathecate clades. Furthermore, the exoskeleton of C. medeae, consisting of a chitin-protein network and a thin exosarc layer, differs from the pseudohydrotheca observed in other hydroids, justifying the use of the term “exoskeleton tube” to describe it.

We thank the National Museum of Natural History, Smithsonian Institution, and its curator, Allen Collins, for providing samples for histological analysis. We also thank José Eduardo A.R. Marian (IB-USP) for his assistance with the histological procedures. We are also grateful to Rizman Idid and the anonymous reviewer for their valuable suggestions and comments on improving the manuscript, and to Izwandy Idris for his assistance during the editorial process. This work is a contribution of the NP-BioMar (USP) and Medusozoa México.

Additional Information and Declarations

Competing Interests

Author Contributions

Data Availability

The authors declare there are no competing interests.

María A. Mendoza-Becerril conceived and designed the experiments, performed the experiments, analyzed the data, prepared figures and/or tables, authored or reviewed drafts of the article, and approved the final draft.

Antonio C. Marques conceived and designed the experiments, analyzed the data, authored or reviewed drafts of the article, and approved the final draft.

The following information was supplied regarding data availability:

The histological slides produced are available in the National Museum of Natural History, Smithsonian Institution’s collection: USNM1003100.

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
