# Peer review of "A morpho-histological analysis of the exoskeleton of Clathrozoella medeae (Cnidaria, Hydrozoa) reveals insights into the taxonomy of Clathrozoellidae and Hydroidolina"

_PeerJ, doi:10.7717/peerj.18407_

## Round 0.1 · original submission · Minor Revisions

All three reviewers unanimously agreed that the manuscript is well written and the science is sufficient for publication in PeerJ. Nonetheless, several areas need improvement to increase the clarity of the content.

I strongly believe that, after addressing the comments by all reviewers, the manuscript will be ready for publication.

Reviewer 1 ·

Basic reporting

The manuscript represents a new and interesting contribution to the knowledge of hydrozoans. The article is written in correct English, but in some cases some corrections have been made to improve the understanding of the text/results (see pdf attached) and some other changes are suggested to make the formatting uniform throughout the text.
Congratulations to the authors for the quality of the figures, they explain very well in each one, to which part of the colony each view belongs to.
Please see some notes for changes/suggestions in section 'Additional comments'.

Experimental design

No comment.

Validity of the findings

No comment.

Additional comments

1. Authors must follow the same format when citing bibliographic works in the text. Sometimes some references are not italicised and others are. Also, sometimes they use & and sometimes and..... You should keep the format consistent.
2. In figure 2k, the arrow is a little misaligned. It seems that it is not pointing to the desmoneme, review this.
3. In the figures, the authors use the abbreviation 'te' to refer to the 'exoskeleton tubes', but in the legends they use 'et', they should use the same abbreviation for both.
4. Finally, a recommendation for formatting ranges of figures, e.g.: A-D, use better the form '–': e.g. A–D

Annotated reviews are not available for download in order to protect the identity of reviewers who chose to remain anonymous.

·

Basic reporting

The manuscript is well written and provide valuable morpho-histological documentation of the exoskeleton of Clathrozoella medeae. The introduction sufficiently furnished the reader with the systematics and taxonomic background of Clathrozoellidae and most of the references were relevant and current. The findings were well presented and discussed appropriately to validate their current taxonomic status.

A lot of work was done to capture such impressive morpho-histological images of the various structures using various staining methods. Figures were of good resolution , and the captions of figures and tables were self explanatory.

In general the manuscript is of good quality and merits publication upon some minor correction.

Experimental design

The research was conducted using appropriate methods and achieved the objectives of the study. It definitely contributes significantly to the taxonomy of taxonomy of Clathrozoellidae and Hydroidolina in general.

In the Material and Methods section , my only suggestion for improvement is to also describe the microscopy method / equipment used after mentioning the various staining methods.

Validity of the findings

The images and findings of this research is invaluable towards the study of hydroids, with very useful documentation and morpho -histological description of the exoskeleton of Clathrozoella medeae.

The discussion was well presented and validated with relevant references, while highlighting the importance of having detailed morpho -histological description for this group of organisms.

Additional comments

This manuscript is well written and merits publication upon some minor corrections:

1) In the Material and Methods section , my only suggestion for improvement is to also describe the microscopy method / equipment used after mentioning the various staining methods.

2) Captions of Figure2 and Figure3 ; instead of 'et' it should be 'te' which corresponds to the 'tubes of exoskeleton'. ' Exoskeleton tubes' should be ' 'tubes of exoskeleton' in Figure 2 caption.

3) Notes of Table 1: Toluidina blue should be toluidine blue

·

Basic reporting

This paper discusses on morphological analyses of the hydroid Clathrozoella medeae. The main contribution of the paper is to elucidate the histology of species with a focus on its exoskeletal structure using various staining techniques. The results add important information on taxonomy and evolutionary relationships of Clathrozoellidae. The English language, references, and article structure are adequate and of an excellent level. The introduction provides a complete historical survey of the taxonomic problems analyzed by different authors over the years. Despite the drawing of the colony in Figure 2, I missed a photo of the colony to better illustrate its morphology and growth pattern. However, in general the figures and table satisfactorily support the text.

Experimental design

The object of study is well defined, and the methodology is consistent with the goals of the study. The methods were described with sufficient detail. The observation of the exoskeleton of the species through histological techniques allowed the authors to make valuable inferences about important taxonomic characters for the species and family. It would have been interesting to do the same histological analysis of specimens from other museums collected in different locations, if there are any, of course. Therefore, it would be interesting to indicate somewhere in the text where the specimens of this species are deposited.

Validity of the findings

The results of this work contribute significantly to the species and family studied, demonstrating that morpho-histological analysis is quite informative for the group, being able to solve historical problems within Hydroidolina. The discussion on the exoskeleton of C. medea and the terminological discussion are important to ratify the position of the species within its family. The conclusions are clear supporting results.

Additional comments

Please correct the spelling of the word "relationships" on line 87; and the abbreviation "et" on legend of Figure 3 (in the figure the abbreviation is "te").

---

## Round 0.2 · accepted · Accept

You have addressed all comments, and I think it is acceptable not to add additional photos (as Felipe Ferreira Campos suggested). So yes, your manuscript is ready for publication, and I look forward to seeing it published!